# A Tolerance Specification Automatic Design Method for Screening Geometric Tolerance Types

**Guanghao Liu** [1,2], **Meifa Huang** [1,*] **and Wenbo Su** [1]

1   School of Mechanical and Electrical Engineering, Guilin University of Electronic Technology, Guilin 541004, China; 1701101005@mails.guet.edu.cn (G.L.)
2   School of Mechanical and Electrical Engineering, Liuzhou Vocational &Technical College, Liuzhou 545005, China
*   Correspondence: hmhmf@guet.edu.cn

**Abstract:** At present, the automatic generation of tolerance types based on rule-based reasoning has an obvious characteristic: for the same assembly feature, tolerance items are recommended that satisfy all feature characteristics, with a large number of recommendations. For this reason, automatically selecting tolerance types and reducing designer autonomy remains a challenging task, especially for complex mechanical products designed using heterogeneous CAD systems. This article proposes a tolerance specification design method for the automatic selection of assembly tolerance types. Based on the construction of a hierarchical representation model of assembly tolerance information with tolerance-zone degrees of freedom (DOFs), a semantic model of geometric tolerance information with tolerance-zone DOFs and a meta-ontology model of assembly tolerance information representation are constructed. Descriptive logic is used to express the attribute relationships between different classes in the assembly tolerance information meta-ontology model, and screening inference rules are constructed based on the mechanism for selecting assembly tolerance types based on tolerance-zone DOFs. On this basis, a process for selecting assembly geometric tolerance types based on the ontology of tolerance-zone DOFs is formed. Finally, the effectiveness and feasibility of this method were verified through examples.

**Keywords:** tolerance specifications; assembly body; screening; degree of freedom; ontology

## 1. Introduction

Computer-aided tolerance design is the key content of the digitalisation of the intelligent manufacturing of products [1,2], in which the realisation of tolerance specification automatic generation (TSAG) is one of the key links. With the efforts of many scholars, the research on achieving the computer-aided design of tolerance specification has also made important progress: Zhang et al. [3] constructed a TSAG algorithm based on a multi-colour set method; Zhang et al. [4] and Qie et al. [5] developed a TSAG method based on a hierarchical tolerance information representation model; and Luo et al. [6] and Johannesson et al. [7] proposed the adjacency matrix method to represent constraint relations of assemblies. On this basis, these methods are combined with computer technology to achieve the automatic reasoning of tolerance types for assembly parts. However, the above methods of automatically generating tolerance types have shortcomings. When reasoning about the tolerance types of the same assembly element (AE) of a part, it will reason about all the shape, orientation and position tolerances that correspond to the characteristics of the AE. However, in the actual detailed design of the tolerance specification scheme for the AE, further manual screening is required based on the geometric functional requirements and working characteristics of the product [4–8]. This situation can increase uncertainty and decision-making difficulties in product development.

In addition, the design of complex products may be carried out using commercial CAD systems with different kernels. In the automatic generation of assembly tolerance types, it

is necessary to implement a data exchange of tolerance specification semantic information between heterogeneous CAD systems [3,9–11]. However, such tolerance specification information is difficult to express on a computer, making it difficult to efficiently generate and filter tolerance types for assembled parts in heterogeneous CAD systems [11,12].

In order to achieve the data exchange of tolerance specification information between heterogeneous CAD systems and achieve the automatic selection of tolerance types, the following two tasks need to be carried out well. One is to select appropriate tolerance type screening methods. The second is to choose an appropriate product data representation language to meet the requirements of the tolerance information representation in computers and exchange between heterogeneous CAD systems [2,11,12].

In the process of designing assembly tolerance (AT) specifications, the use tolerance-zone DOFs can effectively achieve the screening and optimisation of common types and meet the geometric functional requirements of the assembly [13,14]. In the literature [14], our team has analysed the influence of the tolerance-zone DOFs of assembly feature surfaces (AFSs) on the geometric functional tolerance of the assembly in the case of a single datum in a typical assembly and obtained the control parameter DOFs; based on this, we have established the process of manually screening the tolerance types with the control parameter DOFs. In order to reduce the uncertainty of manually screening tolerance types and improve design efficiency, it is necessary to adopt suitable means to achieve the automatic screening of tolerance types. And, in terms of tolerance information data exchange between heterogeneous CAD systems, using OWL (web ontology language) to achieve knowledge representation and tolerance semantic information has a good application effect in tolerance specification design [11,12,15] and inspection [16]. Therefore, this paper proposes an assembly geometry tolerance type screening method based on tolerance-zone DOFs tolerance-band-free ontology, using tolerance-band freedom to achieve the screening of tolerance types and at the same time using ontology technology to achieve the screening of tolerance types of automatic reasoning.

Based on the research, the main organisation of the paper is as follows: Section 2 reviews the related research. Section 3 constructs an improved assembly tolerance representation model containing the tolerance-zone DOF layer according to the meaning of tolerance-zone DOFs and illustrates and defines the contents of the representation model. Section 4 gives a semantic model of geometric tolerance information with tolerance-zone DOFs. On this basis, firstly, the assembly tolerance meta-ontology model for screening tolerance types is constructed; secondly, the inference rules for screening tolerance types are constructed according to the process of screening tolerance types with tolerance-zone DOFs; finally, the assembly tolerance domain ontology framework of the tolerance-zone DOFs is obtained. In Section 5, the tolerance automatic generation process for screening tolerance types is constructed according to the contents of Section 4. Finally, an example study is carried out to verify that the tolerance automatic generation method proposed in this paper can achieve the automatic screening of tolerance types.

## 2. Related Work

In recent years, ontology has been adopted for modelling knowledge related to product design to enable knowledge reuse and information sharing between different applications [16–19]. Also, TSAG is an important area of computer-aided design [12].

### 2.1. Application of Ontology in the Field of Product Design Tolerance Design

Ontologies, as conceptual explicit specifications, are known for their ability to explicitly represent and exchange data semantics [20,21] and play an important role in information sharing, application integration, interoperability implementation and knowledge reuse [18]. They are increasingly being used in the product development process to share data and enable interoperability between heterogeneous product design software [17].

Currently, in order to improve design efficiency and quality, some scholars are trying to use the ontology approach for computer-aided design [22,23] and computer-aided manufac-

turing [24,25]. Many scholars use ontologies to formalise the knowledge of the design process and build an ontology library of knowledge applications in the course of product design and manufacturing; on this basis, they design inference rules to reason out the relevant design results and new knowledge, which improves the design efficiency [18,19,22–26].

Zhong et al. [27] use description logic language to construct a tolerance specification design ontology knowledge base and to design inference rules to achieve tolerance type reasoning. Peng et al. [28] propose a geometric tolerance semantic (GTS) representation model based on GeoSpelling and use OWL to establish an ontology model of tolerance domain knowledge, which achieves the rationality of three-dimensional annotation tolerance information inspection. Sarigecili et al. [29] and Qie et al. [5] established an OWL semantic model of the GT&T and geometric information of a product and implemented an ontology representation of the geometric tolerance information of the parts; based on this, the assembly tolerance analysis of the product was completed.

*2.2. Screening of Automatically Generating AT-Types*

From the preface of Section 1, it can be seen that currently, when using the automatic generation method of tolerance types for the design of tolerance specifications, it is necessary to manually review the tolerance schemes that are obtained by inference and finally determine the annotated tolerances.

Shah et al. [30] and Desrochers et al. [31] combined tolerance design methods based on technology- and topology-related surfaces with computer technology to construct an AT-type automatic generation process. In the literature [32–34], an automatic generation method of the AT type was developed on the basis of assembly positioning constraints. These methods are usually optimised by substitution with human participation. Based on the use of positional tolerance for target feature surface features, relevant directional tolerances are sequentially selected to replace the positional tolerance.

Using the ontology-based approach [5,12,34,35] to infer tolerance types, it is easier to realise the intelligence of automatic tolerance specification generation than when using other methods. Having used this kind of method to obtain the tolerance type, designers often use the following screening methods to determine the final marked tolerance.

One method involves screening based on the intuitiveness of the tolerance symbols. The second is based on the type of tolerance zone, which reflects the comprehensive principle of the tolerance zone. The third is to optimise based on the functional properties of the mating parts. For example, the mating feature surface of a rotary motion axis generally uses the full runout control comprehensive tolerance instead of other directional positional tolerances [10,20]. The fourth in the literature [14] proposes the method of screening AT types using tolerance-zone DOFs, which can effectively reduce the number of tolerances. However, the screening method based on tolerance-zone DOFs needs to consider the representation of tolerance-zone DOFs in different coordinate systems, and it is easy to make mistakes when performing manual calculations of tolerance-zone DOFs. In this respect, it also increases the difficulty of using this method.

In summary, in this paper, we further study the construction of a hierarchical table model of assembly information with a layer of tolerance-zone DOFs based on the method of screening the tolerance types of assemblies using tolerance-zone DOFs. On this basis, we construct a tolerance specification design ontology to realise the automatic screening of the tolerance types of assembly parts.

## 3. Constructing an AT Representation Model Containing Tolerance-Zone DOF Layers

In order to achieve the screening of tolerance types and obtain a smaller number of tolerance types that satisfy the geometric functional requirements of a product, we combine the attribute of tolerance-zone DOFs defined in the literature [14] with the hierarchical AT information representation model to obtain an improved AT information representation model, as shown in Figure 1. The model adds a tolerance-zone layer and a DOF layer to the hierarchical tolerance representation model proposed by Qin et al. [35].

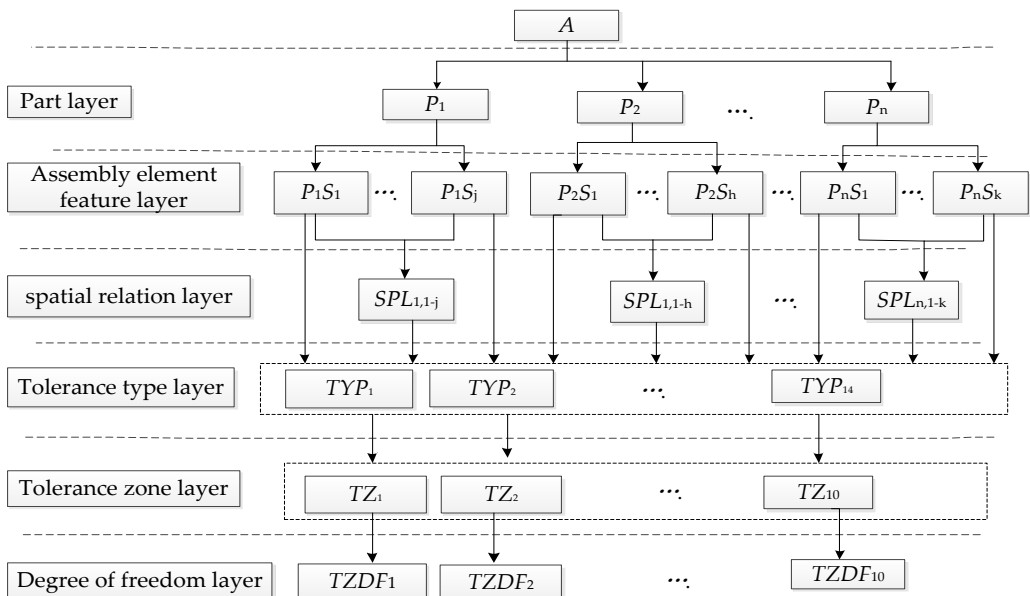

**Figure 1.** Assembly tolerance information representation model. A denotes the assembly, P denotes the part, PS denotes the part assembly feature surface (AFS), SPL denotes the assembly feature element (AFE) spatial relationship, TYP denotes the assembly tolerance type, TZ denotes the tolerance domain and TZDF denotes the tolerance-zone DOFs.

In the improved hierarchical assembly tolerance representation model, the first layer is the part layer, the second layer is the assembly element feature layer, the third layer is the assembly feature element (AFE) spatial relationship layer and the fourth layer is the tolerance type layer. The content and representation of the four layers defined above can be found in the literature [28,36], which summarises the definition and representation of knowledge in the field of tolerance specification design. The content and presentation of the description of the tolerance-zone layer and the tolerance-zone DOF layer are the focus of this paper.

### 3.1. Tolerance-Zone Layer

The tolerance-zone layer specifies the direction of the tolerance zone and the mapping relationship between tolerance types and assembly element feature. They provide the reasoning basis for determining the representation of DOF vectors in tolerance zones. In the case of a single benchmark, the tolerance has nine basic shapes, as shown in Table 1. When applying different tolerance types to different assembly features, the shape of the tolerance zones may vary. The tolerance zone type relationship matrix between the tolerance type (single datum) and the assembly feature type is shown in Table 2. The relationship between them can be represented by the adjacency matrix $M_{T,21 \times 6}$, as indicated in Equation (1).

$$M_{T,21 \times 6} = \begin{bmatrix} C_{1,1} & C_{1,1} & \dots & C_{1,6} \\ C_{2,1} & C_{2,1} & \dots & C_{2,6} \\ \dots & \dots & \dots & \dots \\ C_{21,1} & C_{21,2} & \dots & C_{21,6} \end{bmatrix} \tag{1}$$

where the rows of the matrix $M_{T,21 \times 6}$ represent the tolerance types and the columns correspond to the AFE types, and the values of the elements $C_{i,j}$ are related to the correspondence between the tolerance types and the AFEs, and if the tolerance type of the ith row has a mapping relationship with the type of AFE of the jth row, the value will be the tolerance zone type designation in Table 1, otherwise it will be "0".

**Table 1.** Relationship matrix between the type of tolerance zone and the DOFs.

| Tolerance Code | TS₁ | TS₂ | TS₃ | TS₄ | TS₅ | TS₆ | TS₇ | TS₈ | TS₉ |
|---|---|---|---|---|---|---|---|---|---|
| TZ form |  |  |  |  |  |  |  |  |  |
| DOI [1] | $T_Z$ $R_X, R_Y, R_Z$ | $T_X, T_Y$ $R_X, R_Z$ | $T_X, T_Y$ $R_Z$ | $T_Z$ $R_X, R_Y, R_Z$ | $T_Z$ $R_Z$ | $T_Z$ $R_Z$ | $R_X, R_Y, R_Z$ | $T_Y$ $R_X, R_Z$ | |
| DOFs | $T_X, T_Y$ | $T_Z,$ $R_Y$ | $T_Z,$ $R_X, R_Y$ | $T_X, T_Y,$ | $T_X, T_Y,$ $R_X, R_Y,$ | $T_X, T_Y$ $R_X, R_Y$ | $T_X, T_Y, T_Z$ | $T_X, T_Z$ $R_Y,$ | $T_X, T_Y, T_Z$ $R_X, R_Y, R_Z$ |
| Representation | Ti (1,1,0) Ri (0,0,0) | Ti (0,0,1) Ri (0,1,0) | Ti (0,0,1) Ri (1,1,0) | Ti (1,1,0) Ri (0,0,0) | Ti (1,1,0) Ri (1,1,0) | Ti (1,1,0) Ri (1,1,0) | Ti (1,1,1) Ri (0,0,0) | Ti (1,1,1) Ri (0,1,0) | Ti (1,1,1) Ri (1,1,1) |

[1] DOI represents degree of invariance.

**Table 2.** Mapping matrix of tolerance zone type relationship between assembly features and tolerance types.

| Tolerance Number | Tolerance Type | Common Typical Assembly Feature Features (Exported Features) | | | | | |
|---|---|---|---|---|---|---|---|
| | | Cylindrical Surface (Line) | Plane (Plane) | Free-form Surface (Point, Line, Surface) | Sphere (Point) | Circular Countertop (Point, Line) | Prismatic Surface (Line, Surface) |
| TT01 | ○ | TS₄ | | | TS₄ | TS₄ | |
| TT02 | Single direction — | TS₂ | TS₂ | | | TS₂ | TS₂ |
| TT03 | In any direction — | TS₅ | | | | | |
| TT04 | ▱ | | TS₃ | | | | |
| TT05 | ⌀ | TS₆ | | | | | |
| TT06 | ⌒ | | | TS₈ | | TS₈ | |
| TT07 | ⌓ | | | TS₉ | | TS₉ | |
| TT08 | Single direction // | | TS₃ | | | | |
| TT09 | In any direction // | TS₅ | TS₃ | | | | |
| TT10 | Single direction ⊥ | TS₃ | | | | | |
| TT11 | In any direction ⊥ | TS₅ | TS₃ | | | | |
| TT12 | In any direction ∠ | TS₅ | TS₃ | | | | |
| TT13 | Single directional ∠ | TS₃ | | | | TS₂ | |
| TT14 | ≡ | | | | | | TS₃ |
| TT15 | ◎ | TS₃ | | | | | |
| TT16 | ◎ | TS₁ | | | TS₁ | | |
| TT17 | In any direction ⊕ | TS₅ | TS₃ | | TS₇ | | |
| TT18 | Radial circular ↗ | TS₄ | | | | | |
| TT19 | Axial circular ↗ | | TS₂ | | | | |
| TT20 | Total radial ↗↗ | TS₆ | | | | | |
| TT21 | Total axial ↗↗ | | TS₃ | | | | |

TS₁, TS₂…TS₉—see Table 1 for tolerance-zone marking codes. Symbol // indicates parallelism, symbol ⊥ indicates perpendicularity, symbol ◎ indicates coaxiality, t, symbol — indicates straightness, symbol ⊕ indicates positionality, symbol ⌀ indicates cylindricity, symbol ○ indicates roundness, symbol ▱ indicates flatness, symbol ⌒ indicates line profile, symbol ⌓ indicates surface profile, symbol ≡ indicates symmetry, symbol ↗ indicates run-out, symbol ∠ indicates inclination and symbol ↗↗ indicates total run-out.

### 3.2. Tolerance-Zone DOF Layer

The literature [14] defines tolerance-zone DOFs. The relationship matrix between the type of tolerance zone and the DOFs is shown in Table 1. The main content of this layer is to determine the element values of the tolerance-zone DOF vector in different reference systems. We represent the DOF vector of the tolerance zone as $V$. If the coordinate system of the assembly is different from the coordinate system marked by each tolerance zone in Table 1, the element values in vector $V$ will change. The mapping relationship between the

DOFs of each tolerance zone and the coordinate direction is represented by the adjacency matrix shown in Equation (2):

$$M_{vi,3\times6} = \begin{bmatrix} x\_Txi & x\_Tyi & x\_Tzi & x\_Rxi & x\_Rxi & x\_Rxi \\ y\_Txi & y\_Tyi & y\_Tzi & y\_Rxi & y\_Ryi & y\_Rzi \\ z\_Txi & z\_Tyi & z\_Tzi & z\_Rxi & z\_Ryi & x\_Rzi \end{bmatrix} = \begin{bmatrix} X\_DOF \\ Y\_DOF \\ Z\_DOF \end{bmatrix} \quad (2)$$

In the matrix $M_{vi,3\times6}$, $i$ $\{i = X, Y, Z\}$ represent the tolerance-zone DOF vectors for the principal orientation of the coordinate system (usually the normal orientation of the tolerance feature), such as **X_DOF** for the X axis, **Y_DOF** for the Y axis and **Z_DOF** for the Z axis. As shown in Table 1, for tolerance band TS5, if the principal direction of the tolerance band is the Z axis, the DOF vector **V = Z_DOF** is (1 1 0 1 1 0). (Note: when the coordinate system is changed, if the Z-axis direction is converted to the X-axis direction, the X axis is the main direction; if the Z-axis direction is converted to the Y-axis direction, the Y axis is the main direction).

## 4. Construction of an Ontology Framework for Assembly Tolerance Domain Oriented to Tolerance-Zone DOFs

A complete ontology contains classes, properties and instances, where classes are the canonical and clear descriptions of domain concepts, properties are the descriptions of the characteristics of the concepts and instances are the dissimilar individuals contained in the concepts. An OWL ontology based on description logic can use TBox (terminology box) to represent classes and attributes and use ABox (assertions box) to represent instances [36]. Section 4 builds a tolerance knowledge domain ontology by combining a geometric tolerance information representation model with tolerance-zone DOFs with a hierarchical assembly tolerance representation model to realise the semantic representation of assembly tolerance information.

### 4.1. Semantic Modelling of Geometric Tolerance Information with Tolerance-Zone DOFs

According to ISO 1101 and the new-generation GPS [36], geometric tolerance components include three types of feature information: geometric tolerance objects, tolerance types, and tolerance zones. In the literature [14], the process of screening the tolerance types of the AFE in a part is constructed by defining the tolerance-zone DOFs and defining the control parameter DOFs of the assembly, and thus the screening of tolerance types is thus realised. Therefore, based on the engineering semantics of tolerance information, a semantic representation model for geometric tolerances based on tolerance-zone DOFs is constructed, as shown in Figure 2.

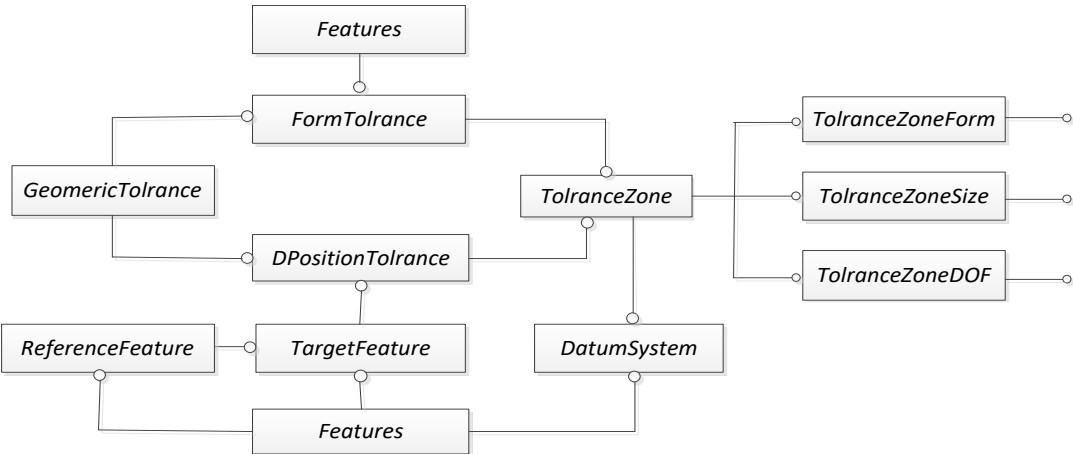

**Figure 2.** Semantic representation model for geometric tolerance.

The rectangles in Figure 2 represent entities, and the lines connecting the rectangles represent the relationships between the entities (called object attributes). The rectangles connected by small circles represent the value domain of the attribute, and the entities connected by no small circles represent the definition domain of the attribute. The value domains of the data attributes are of various data types.

### 4.2. Construction of an AT Information Meta-Ontology Model Based on Tolerance-Zone DOFs

Combining the GTS representation model with the tolerance-zone DOF attribute and the hierarchical AT information representation model [35,36], we constructed the AT information meta-ontology model, as shown in Figure 3, where, Assembly denotes assembly, Part denotes the part, AssemblyFeature denotes AFE, IdealFeature denotes an ideal feature element, RealFeature denotes the actual AFE, TargetFeatures denotes the target feature element, ADF stands for assembly positioning connection, ConstrainedFeature denotes constrained feature elements, GeomericTolrance denotes the geometric tolerance, TolranceZone denotes the tolerance domain, Direction denotes the direction of the tolerance domain, ToleranceZoneDOF denotes the tolerance-zone DOFs and ToleranceType denotes the tolerance type. These are the base classes for the ontology for the representation of the AT information.

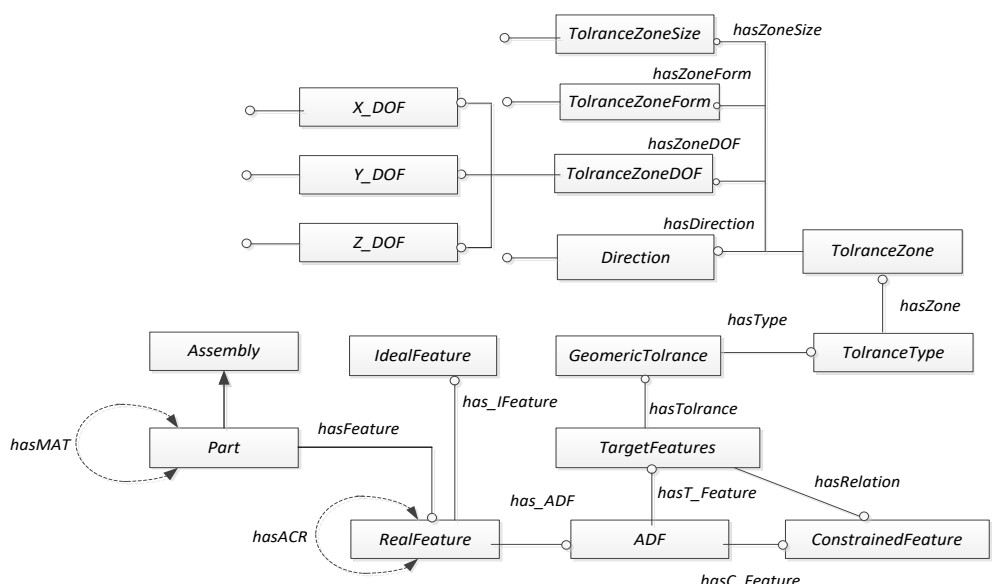

**Figure 3.** Meta-ontology model for assembly tolerance information representation.

### 4.3. Constructing the Class of AT Information Ontology Knowledge Base

Based on the AT design domain knowledge and the assembly precision information meta-ontology model, the conceptual definition of the classes is performed, and the hierarchical relationship diagram of the classes is constructed, as shown in Figure 4. For example, the geometric assembly features are defined as Equation (3), and the conceptual definition of tolerance zones is defined as Equation (4).

$$
\begin{aligned}
&\text{RdealFeature} \equiv \text{RRevolute} \lor \text{RPlane} \lor \text{RSpherical} \lor \text{RCylindrical} \lor \text{RHelical} \lor \text{RPrismatic} \lor \text{RComplex} \\
&\text{RRevolute} \equiv \text{InRR} \lor \text{OutRR}, \text{RPlane} \equiv \text{RPL}, \text{RSpherical} \equiv \text{InSP} \lor \text{OutSP}, \\
&\text{RCylindrical} \equiv \text{InCY} \lor \text{OutCY}, \text{RHelical} \equiv \text{InHE} \lor \text{OutHE}, \\
&\text{RPrismatic} \equiv \text{InPR} \lor \text{OutPR}, \text{RComplex} \equiv \text{InCO} \lor \text{OutCO}
\end{aligned}
\tag{3}
$$

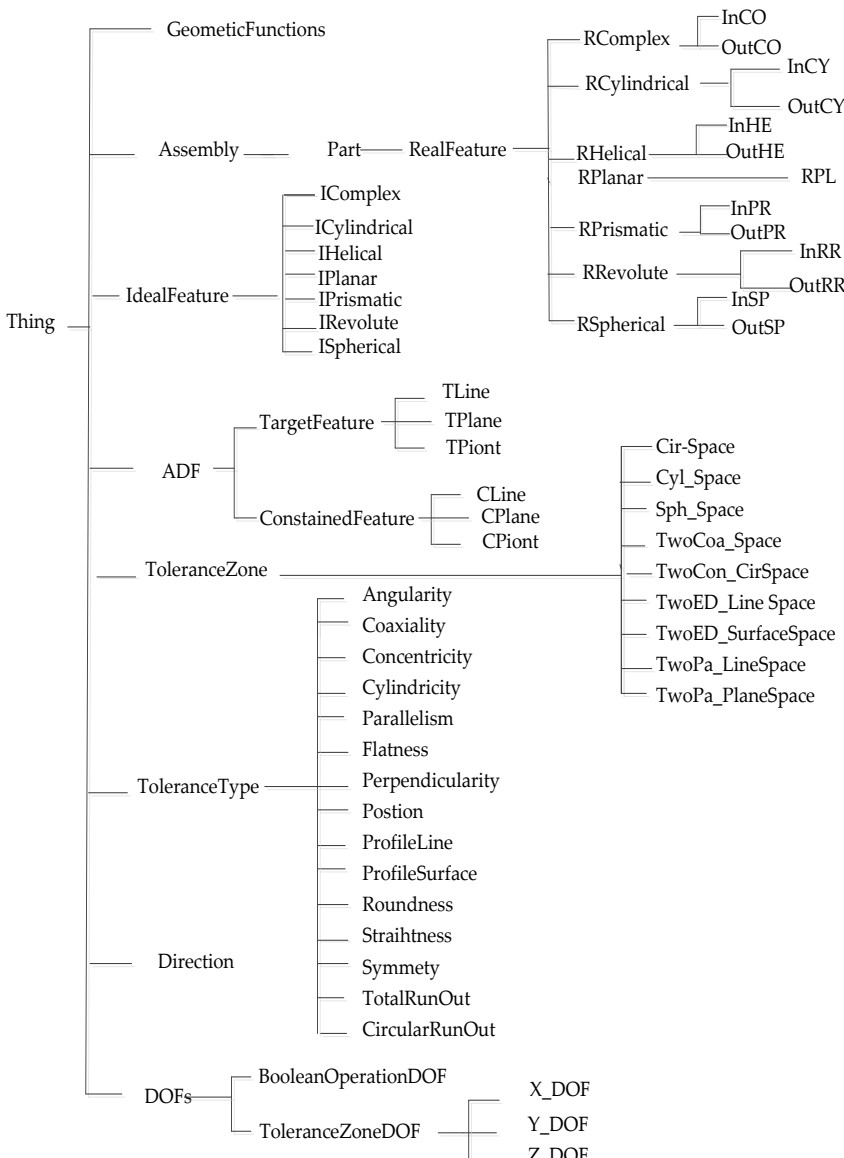

**Figure 4.** Class in meta-ontology for assembly tolerance representation.

### 4.4. Defining Class Properties

The properties of a class are used to describe the relationship between the class and the class for which the assembly tolerance specification is created or to define what kind of characteristics the attribute values of the class have, and they are categorised into two types: object properties and data properties. Based on the hierarchical representation model of AT information in Section 3, we constructed the object attribute hierarchy of the class, as shown in Figure 5; similarly, the data attribute hierarchy of the class was obtained, as shown in Figure 6.

$$
\begin{aligned}
\text{ToleranceZone} \equiv{}& \text{CircularSpace} \lor \text{CylindricalSpace} \lor \text{SphericalSpace} \\
& \lor \text{TwoCoaxialCylSpace} \lor \text{TwoConcentricCirSpace} \lor \text{TwoEqualDistantLine Space} \\
& \lor \text{TwoEqualDistantSurfaceSpace} \lor \text{TwoParalleLine Space} \\
& \lor \text{TwoParallePlaneSpace}
\end{aligned}
\tag{4}
$$

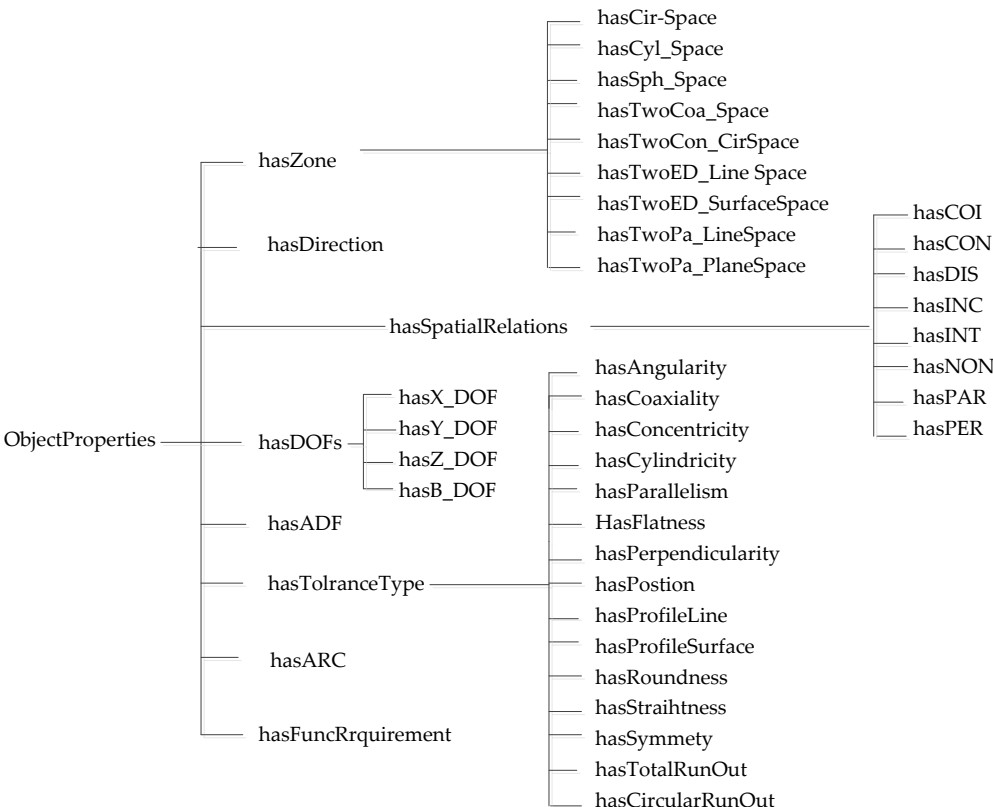

**Figure 5.** Attribute hierarchy of object property of the classes.

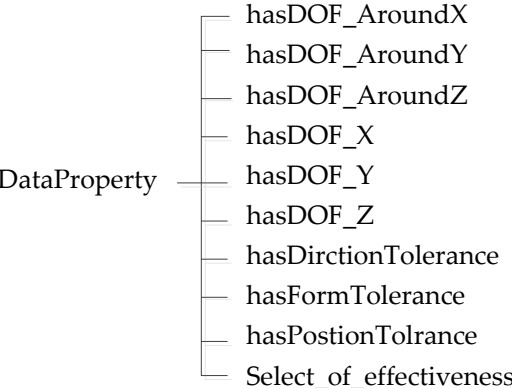

**Figure 6.** Attribute hierarchy of data property of the classes.

The object property is used to describe the binary relationships between classes. For example, the object attribute hasACR indicates that there is a fit relationship between parts, the existing $a_u$ and $a_v$ are the individuals of the parts; if $u \neq v$ and there is an attribute hasACR between $a_u$ and $a_v$, then there is an assertion that Part($a_u$) and Part($a_v$) and hasACR($a_u$, $a_v$) hold.

The datatype attribute defines the internal properties of the class; for example, the hasDOF_AroundX data attribute represents the data attribute of the individual DOF instances of the tolerance-zone DOF vector class DOFs around the X-axis, and the value field of the data attribute is a logical value. Other important data attribute types are shown in Table 3.

After completing the above work, an instance is created, and the specific work of this step will be explained in the case implementation (Section 5.2).

**Table 3.** Domain and range domains of some target and data attributes.

| Serial Number | Object Properties | Domain | Range |
|---|---|---|---|
| 1 | hasZone | RealFeature | ToleranceZone |
| 2 | hasSpatialRelations | ConstainedFeature/RealFeature | TargetFeature/RealFeature |
| 3 | hasDOFs | ToleranceType | DOFs |
| 4 | hasToleranceType | TargetFeature/RealFeature | TolerranceType |
| 5 | hasADF | ConstainedFeature | TargetFeature |
| 6 | HasACR | RealFeature | RealFeature |
| 7 | hasFuncRequirement | Assembly | FuncRequirement |
|   | Data Properties | Domain | Data Types |
| 8 | hasDOF_AroundX | DOFs | Boolean |
| 9 | hasDOF_AroundY | DOFs | Boolean |
| 10 | hasDOF_AroundZ | DOFs | Boolean |
| 11 | hasDOF_X | DOFs | Boolean |
| 12 | hasDOF_Y | DOFs | Boolean |
| 13 | hasDOF_Z | DOFs | Boolean |
| 14 | hasDirection | ToleranceZone | String |
| 15 | Select_of_effectiveness | TolerranceType | Boolean |

*4.5. OWL/SWRL Representation for AT Information Based on Tolerance-Zone DOFs*

How the meta-ontology for representing AT information based on tolerance-zone DOFs implements the OWL/SWRL representation requires the selection of an appropriate tool [30]. Protégé 5.5 is a commonly used tool for implementing ontology creation that provides an integrated environment for visually creating, editing and saving ontology. It facilitated the creation of the classes and subclasses shown in Figure 3 as well as the specification of the class attributes to form the ontology-based terminology concept set TBox; it also facilitated using the SWRL rule language to describe the mapping relationships shown in Table 2. It is also convenient to apply the SWRL rule language representation to the correspondences between tolerance types and AFE types and between tolerance types and spatial relations [12,35]. The above relationships form the basis for the construction of inference rules for the automatic generation of tolerance types and the automatic screening of tolerance types.

4.5.1. SWRL Representation of Generation Rules for AT Specifications

Tolerance types are usually classified into two main categories: shape tolerance and directional positional tolerance. The inference rules for tolerance type generation are constructed based on the classes and object properties and data attributes built by the protégé tool.

1. SWRL representation of rules for generating shape tolerance types. The form tolerance type is related to the target feature element type. SWRL rules are constructed according to the correspondence between the type of the target feature element and the form tolerance [3], and the automatically generated SWRL rule base for form tolerance items can be obtained [5,23,31]. For example, the commonly used shape tolerance types for cylinders are cylindricity, centre element axis straightness, cross-section roundness and rotation bus straightness; the inference rules are as follows: $RCyindrical(?x) \wedge ICyindrical(?y) \wedge hasNON(?x,?y) \rightarrow hasCylindricity(?x,?y) \wedge HasRoundness(?x,?y) \wedge hasStraihtness(?x,?y)$.

2. SWRL representation of rules for generating orientation–position tolerance types. The orientation–position tolerance type of the target feature element is closely related to the orientation and position of the constrained feature element. The mapping relationship between the assembly tolerance type and the spatial relationship is expressed as the corresponding SWRL inference rule, and the SWRL rule base for the automatic generation of the orientation–position tolerance type can be established. For example, if the target feature plane and the constraint feature plane are perpendicular to each

other, the tolerance types that can be selected for the target feature elements are positional and perpendicular. The inference rules are expressed as follows: TPlane(?x) ∧ CPlane(?y) ∧ hasPER(?x,?y)→ hasPerpendicularity(?x,?y) ∧ hasPostion(?x,?y).

4.5.2. SWRL Representation of Inference Rules for Screening Tolerance Types

Section 3 defines the adjacency matrix $M_{vi,3\times6}$ for the tolerance-zone DOF layer and the mapping matrix $M_{T,21\times6}$ for the tolerance zone type layer. The tolerance type selection algorithm using the tolerance-zone DOFs proposed in reference [14] is shown in Figure 7. The protege 5.5 tool is used to complete the instantiation of different tolerance-zone DOF vectors for different tolerance types of the same AFE. Based on this, the SWRL rule language is used to describe the tolerance type selection algorithm using tolerance-zone DOFs, and the SWRL representation of the intelligent tolerance type selection rules is constructed.

1.  Reasoning rules for common DOF vector $V_c$. In the coordinate system, if the normal direction of the target feature element is the Y-axis direction, the vector $V_t$ is an instance of the class Y_DOF. The measurement reference feature DOF vector $V_f$ is an instance of class DOFs. Then, the values of the elements in the vector $V_C$ are the Boolean values between $V_t$ and $V_f$. Based on the adjacency matrix $M_{vi,3\times6}$, which represents the tolerance-zone DOF vectors, the inference rule for finding the values of the six data attributes of the vector $V_c$ is constructed. For example, the SWRL of the inference rule for finding the value of translational DOFs in the Y-axis direction for the vector $V_c$ is expressed as: hasDOF_Y($V_t$,?x) ∧ hasDOF_Y($V_f$,?y) ∧ swrlb:notEqual(?x, ?y) -> hasDOF_Y($V_c$, false); hasDOF_Y($V_t$,?x) ∧ hasDOF_Y($V_f$,?y) ∧ swrlb:Equal(?x, ?y) -> hasDOF_Y($V_c$, ture).
    Similarly, the inference rules can be constructed for the other five data attribute values in the common DOF vector $V_c$.

2.  Reasoning rules for the control parameter DOF vector $V_0$. If $V_p$ is the auxiliary DOF vector, the control parameter DOF vector $V_0$ is the result of the Boolean of the auxiliary DOF vector $V_P$ and the common DOF vector $V_C$. With reference to the method of determining the inference rule for the common DOF vector, the inference rule for determining the six data values in the control parameter DOF vector $V_0$ is constructed. As an example, the SWRL of the inference rule for finding the rotational DOF value of the DOF vector $V_0$ around the Y axis is expressed as follows: hasDOF_aroundY($V_c$,?x) ∧ hasDOF_aroundY($V_p$,?y) ∧ swrlb:notEqual(?x, ?y) -> hasDOF_aroundY($V_0$, false); hasDOF_aroundY($V_c$,?x) ∧ hasDOF_aroundY($V_p$,?y) ∧ swrlb:Equal(?x, ?y) -> has-DOF_aroundY($V_0$, true).
    Similarly, the inference rules can be constructed for the other five data values in the vector of control parameter DOFs $V_0$.

3.  Inference rules for determining the comparative DOFs $V'_{ij\_k}$. If $V_{ij\_k}$ is a vector of DOFs for the different tolerance types of an AFE (subscript *i* is the part number, *j* is the contact surface number, and *k* is the tolerance mark sequence number) and $V_0$ is a vector of control parameter DOFs, then the comparative DOFs vector $V'_{ij\_k}$ is the result of a Boolean operation between $V_{ij\_k}$ and $V_0$. With reference to the method of determining the inference rule for the common DOF vector, the inference rule for determining the values of the six data attributes in the comparative DOF vector $V'_{ij\_k}$ is constructed. As an example, the SWRL of the inference rule for determining the value of the translational DOF in the Y axis of the comparative DOF vector $V'_{ij\_k}$ is ex-pressed as follows: hasDOF_Y($V_{ij\_k}$,?x) ∧ hasDOF_Y($V_0$,?y) ∧ swrlb:notEqual(?x, ?y) -> hasDOF_Y($V'_{ij\_k}$, false); hasDOF_Y($V_{ij\_k}$,?x) ∧ hasDOF_Y($V_0$,?y) ∧ swrlb:Equal(?x, ?y) -> hasDOF_Y($V'_{ij\_k}$, true).

4.  Reasoning rules for screening tolerance types. If the comparison DOF vector $V'_{ij\_k}$ and the control parameter DOF vector $V_0$ are equal, then $V'_{ij\_k}$ is the comparison DOF for the preferred tolerance type. For this purpose, the SWRL rule for filter-ing the tolerance type is obtained and described as follows: hasDOF_X($V_{ij\_k}$,?b) ∧ hasDOF_X($V_0$,?c) ∧ swrlb:Equal(?b, ?c) ∧ hasDOF_Y($V'_{ij\_k}$,?d) ∧ hasDOF_Y($V_0$,?e)

$\wedge$ swrlb:Equal(?d, ?e) $\wedge$ hasDOF_Z $(V'_{ij\_k},?f)\wedge$ hasDOF_Y$(V_0,?g)$ $\wedge$ swrlb:Equal(?f, ?g) $\wedge$ hasDOF_aroundX$(V'_{ij\_k},?k)$ $\wedge$ hasDOF_aroundX$(V_0,?l)$ $\wedge$ swrlb:Equal(?k,?l) $\wedge$ hasDOF_aroundY$(V'_{ij\_k},?m)$ $\wedge$ hasDOF_aroundY$(V_0,?n)$ $\wedge$ swrlb:Equal(?m,?n) $\wedge$ has-DOF_aroundZ$(V'_{ij\_k},?j)$ $\wedge$ hasDOF_aroundZ$(V_0,?k)$ $\wedge$ swrlb:Equal(?j, ?k)->Select_of_effectiveness$(V'_{ij\_k}$,ture).

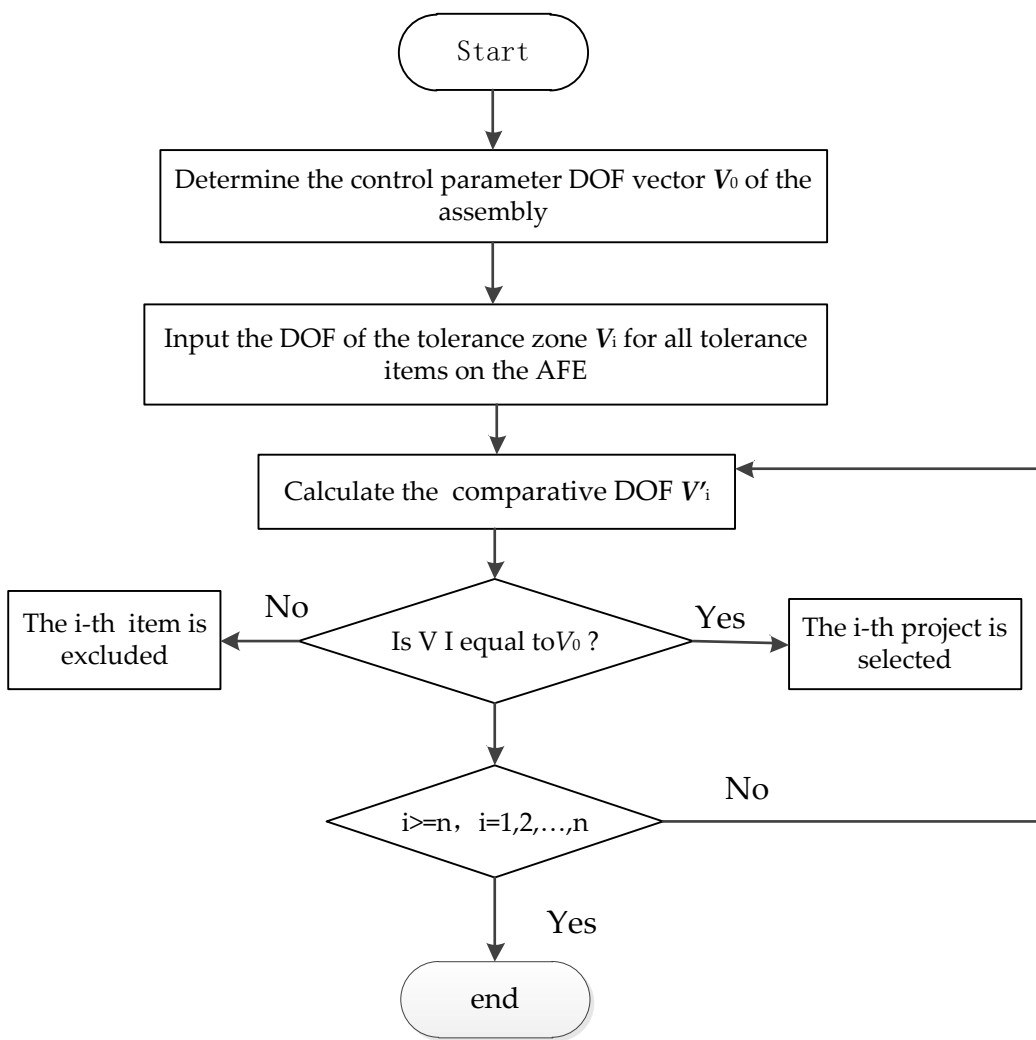

**Figure 7.** Tolerance type screening process [14].

## 5. Implementation and Examples

### 5.1. Constructing an Automatic Generation Algorithm for Screening AT Types Based on Tolerance-Zone DOF Ontology

As can be seen from the content in Section 2, in the process of designing tolerance specifications for assembled workpieces, realising automatic screening of tolerance types is a very important research work after automatic generation of tolerance types. Therefore, we combined the research results from the literature [14,22] and the contents of Sections 3 and 4 to improve the tolerance design process of the assembly and to construct a tolerance specification design process that uses the tolerance-zone DOFs to automatically screen the geometric tolerance types of the assembly, as shown in Figure 8.

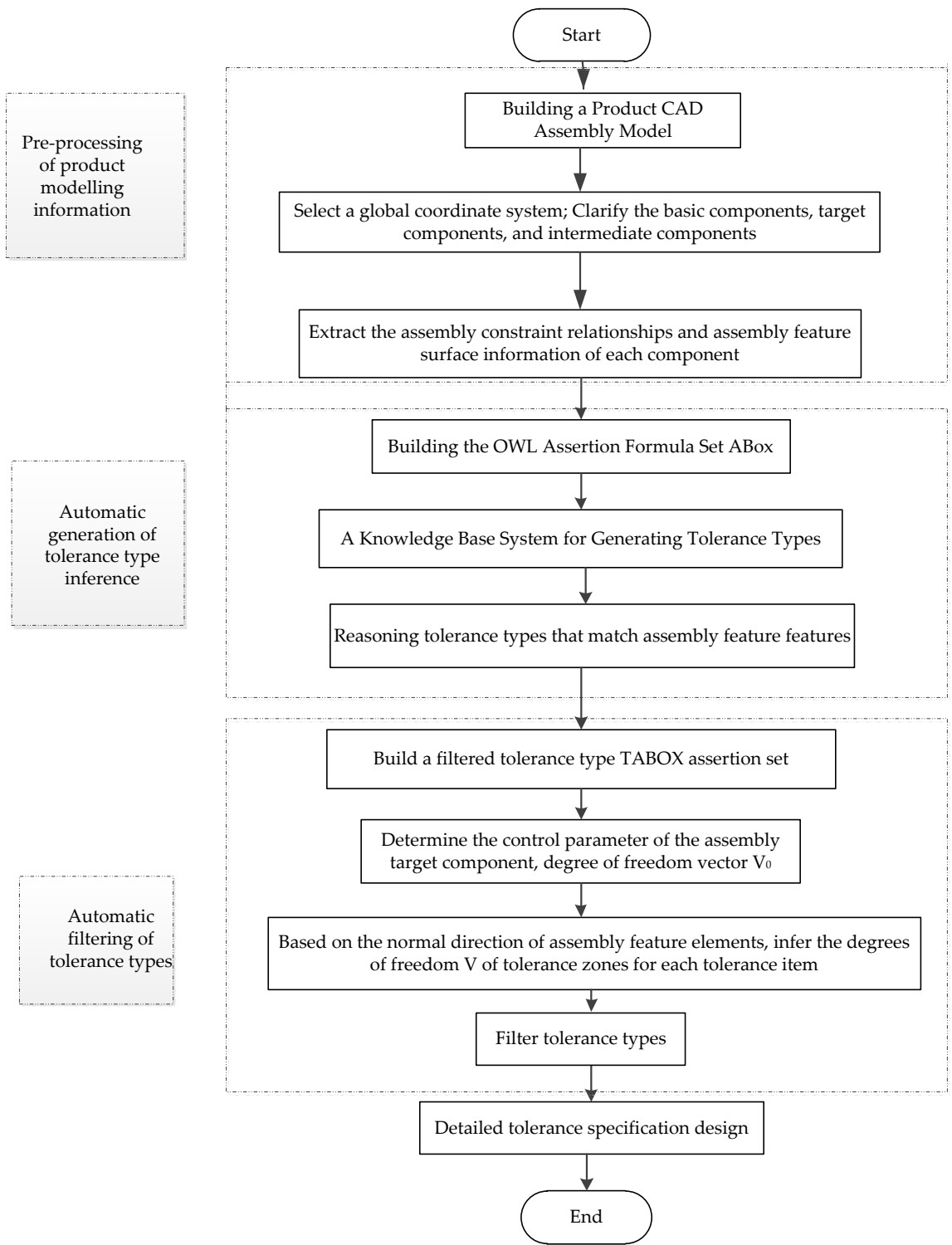

**Figure 8.** Tolerance type screening process.

## 5.2. Example Verification

Figure 9a shows a simplified assembly model of a mechanical product with a geometric tolerance requirement of perpendicularity. Part Part$_4$, which performs linear up and down motion, is actively connected to a hydraulic cylinder drive device. In this section, the workpiece Part$_3$ of the product model was selected as the research object to verify the feasibility of the tolerance specification design process shown in Figure 8.

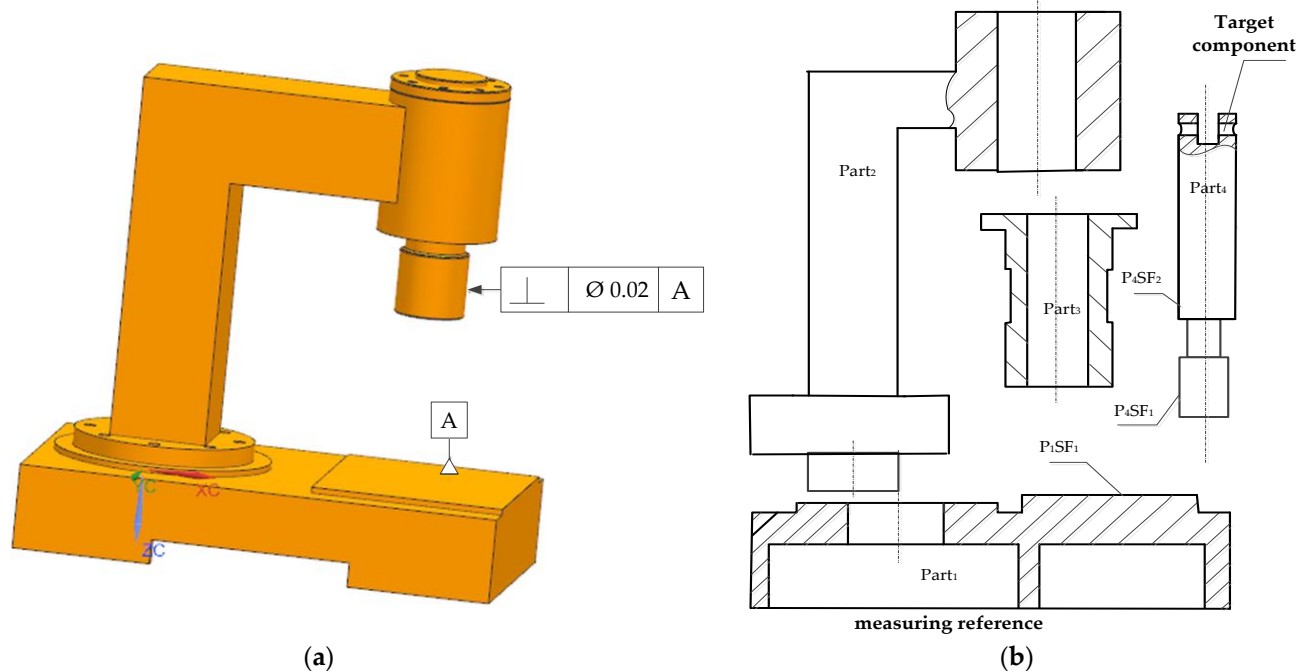

(**a**)  (**b**)

**Figure 9.** (**a**) Three-dimensional assembly model; (**b**) assembly part and coordinates.

5.2.1. Pre-Processing of Product Modelling Information

1. Acquisition of 3D models of products. The CAD 3D assembly model of the product was constructed, as shown in Figure 9a.

2. The measurement target and reference object of the assembly and the global coordinate system were defined. The geometric functional requirement of the assembly was perpendicularity, as shown in Figure 9a. Then, the part $Part_4$ was defined as the target part of the assembly, and Part1 was defined the measurement datum part. According to the measurement elements and datum elements, the global coordinate system was established, as shown in Figure 9b.

3. According to the method introduced in reference [27], we extracted the assembly constraint information of the simple stamping model from the product assembly design data structure in the CAD system, such as the assembly constraint relationship between parts shown in Figure 10. In the figure, $M_{i(Pu,Pv)}$ indicates the assembly relationship between the assembled workpieces, subscript $i$ indicates the assembly number, $Pu,Pv$ indicates the assembled part number, respectively, and $u$ is not equal to $v$. $APJ_{k,l-n}$ indicates the localisation constraint connection between the assembled feature surfaces of the parts, subscript $k$ indicates the assembly number of the parts and $l,m$ indicates the feature surface number of the different parts, respectively; we extracted the assembly feature surfaces (AFSs) information of each part, such as the geometric and coordinate feature information of these AFSs and the geometric spatial relationship between each AFE of the part.

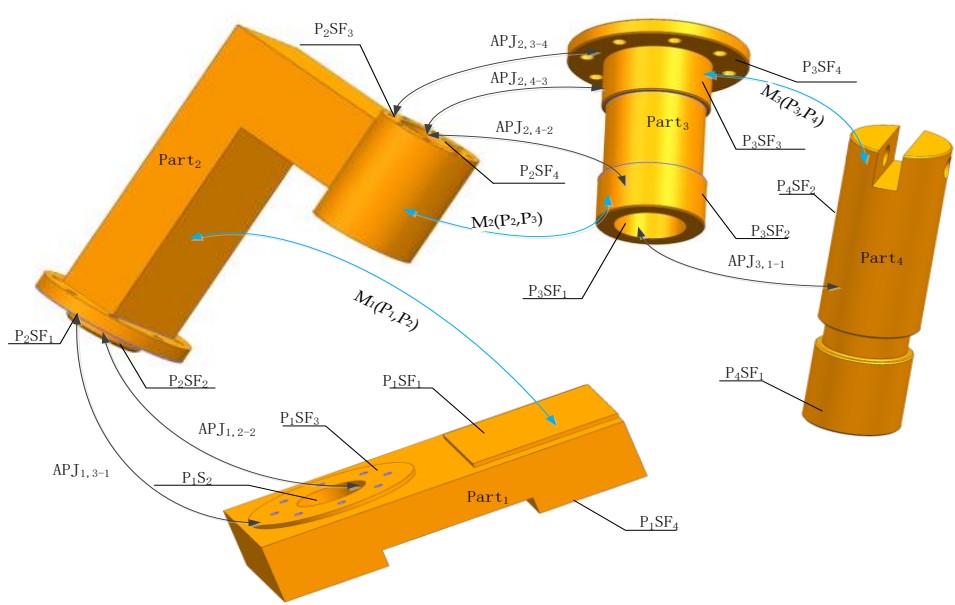

**Figure 10.** The assembly constraint relationships between geometric product components.

5.2.2. Automatically Generate Tolerance Items for Parts of the Assembly

1. Building the axiom set of the OWL assertions ABox. The OWL assertion axiom set ABox was established, which represents the assembly constraint relationships between the geometric product components based on these AFSs of each component and these constraint relationships between features. Taking $Part_3$ in Figure 10 as an example, the constraint relationship assertion AP between the parts was obtained, and AP can be expressed as per Equation (5).

$$A_P = \{Part_2, Part_3, Part_4, hasMAT(Part_2, Part_3,), hasMAT(Part_3, Part_4)\} \tag{5}$$

$A_F$ can be expressed as Equation (6).

$$A_F = \{Cylindrical\ (P_2SF_4),\ Planar\ (P_2SF_3),\ Planar\ (P_3SF_4),\ Cylindrical\ (P_3SF_3), \\ Cylindrical\ (P_3SF_2),\ Cylindrical\ (P_3SF_1),\ Cylindrical\ (P_4SF_2)\} \tag{6}$$

For the assertion of constraints between feature surfaces in part assembly AFR, AFR can be expressed as per Equation (7).

$$AFR = \{ADF_{2,4\text{-}2},\ ADF_{2,4\text{-}3},\ ADF_{2,3\text{-}4},\ ADF_{3,1\text{-}1},\ hasCOI\ (P_2SF_3,\ P_3SF_4), \\ hasCOI\ (P_2SF_4,\ P_3SF_2),\ hasCOI\ (P_2SF_4,\ P_3SF_3),\ hasPER\ (P_2SF_3,\ P_3SF_2), \\ hasPER\ (P_2SF_3,\ P_3SF_3),\ HasCOI\ (P_3SF_1,\ P_4SF_2),\ hasCOI\ (P_3SF_1,\ P_3SF_2), \\ hasCOI\ (P_3SF_1,\ P_3SF_3),\ hasCOI\ (P_3SF_2,\ P_3SF2),\ hasPER\ (P_3SF_1,\ P_3SF_4), \\ hasPER\ (P_3SF_2,\ P_3SF_4),\ hasPER\ (P_3SF_1,\ P_3SF_4)\} \tag{7}$$

2. A knowledge ontology library for the automatic generation of tolerance types was built. Following 4.5.3 and 4.5.4, the input of TBox, a set of terms for the tolerance specification design concepts, was realised using the protege5.5 ontology editor application, as shown in Figure 11. Taking part $Part_3$ as an example, the $A_P$ $A_F$ and $A_{FR}$ assertion sets were input into the ontology editing tool software Protege 5.5 to realise the establishment of the OWL assertion axiom set ABox, and the tolerance type auto-generation knowledge base was finally obtained. The object property assertion of the assembly feature face $P_3SF_1$ of part P3 is shown in Figure 12.

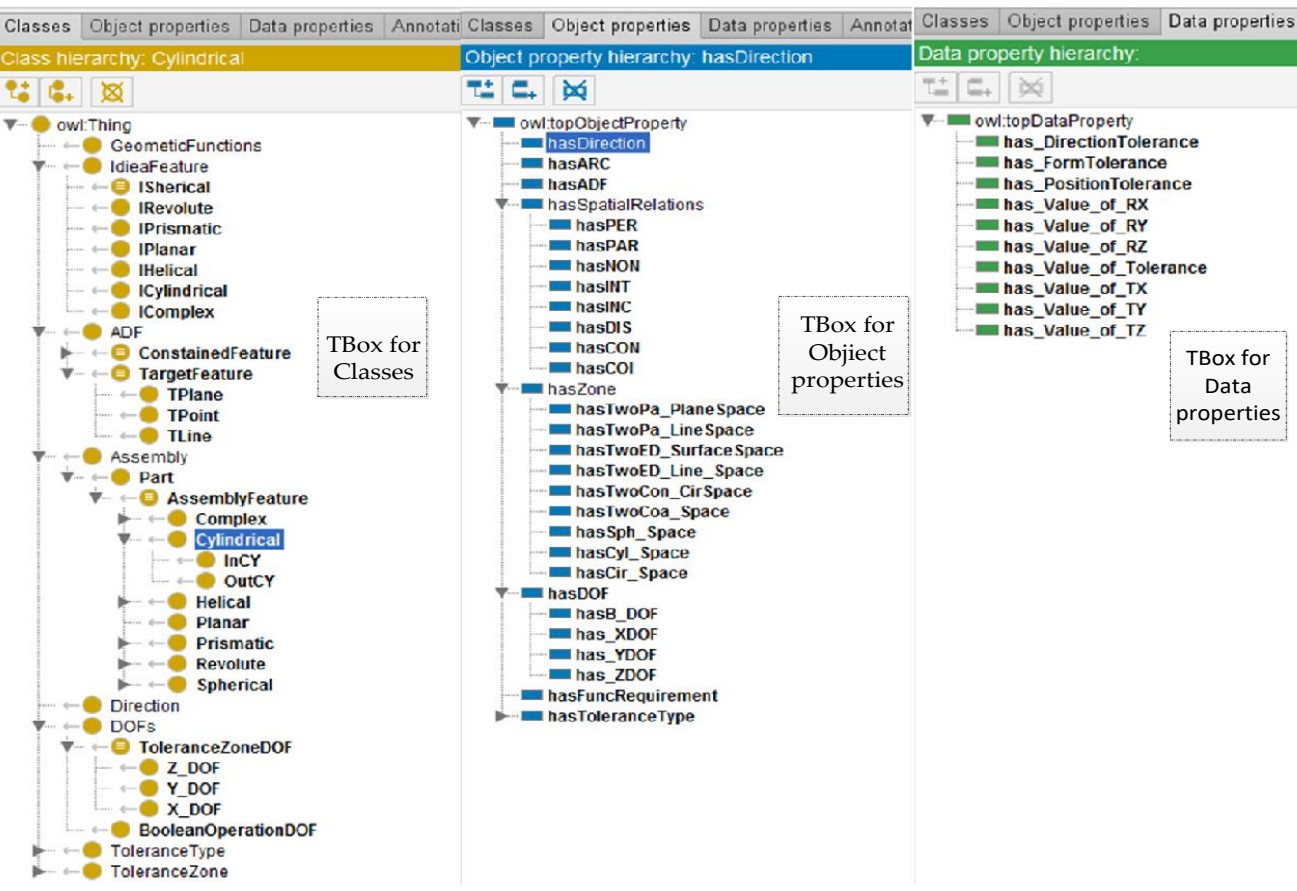

**Figure 11.** Terminology assertion TBox.

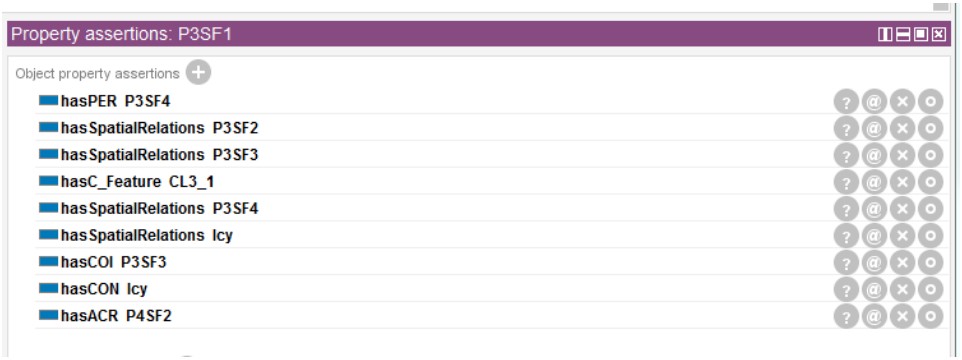

**Figure 12.** Object properties of $P_3SF_1$.

3. Inference generation for tolerance types. The SWRL inference rules [35] were input to automatically recommend tolerance items for each part of the assembly. Taking part $Part_3$ as an example, the inference result of feature surface $P_3SF_1$ was obtained, as shown in Figure 13. The tolerance types for other assembly feature surfaces are shown in Table 4.

**Table 4.** Recommended tolerances for assembly feature faces of part Part$_3$.

| | ASF [1] | In the Plan | In Any Direction | ○ | ⌀ | ↗ | ⤢ | ⊕ | ◎ | // | ⊥ | ▱ |
|---|---|---|---|---|---|---|---|---|---|---|---|---|
| Recommended tolerances | P$_3$S$_1$ | | ✓ | ✓ | ✓ | | | | ✓ | ✓ | ✓ | ✓ | |
| | P$_3$S$_2$ | ✓ | ✓ | ✓ | ✓ | | ✓ | ✓ | ✓ | ✓ | ✓ | ✓ | |
| | P$_3$S$_3$ | ✓ | ✓ | ✓ | ✓ | | ✓ | ✓ | ✓ | ✓ | ✓ | ✓ | |
| | P$_3$S$_4$ | ✓ | | | | | ✓ | ✓ | ✓ | | | ✓ | ✓ |

[1] AFS represents assembly feature surface.

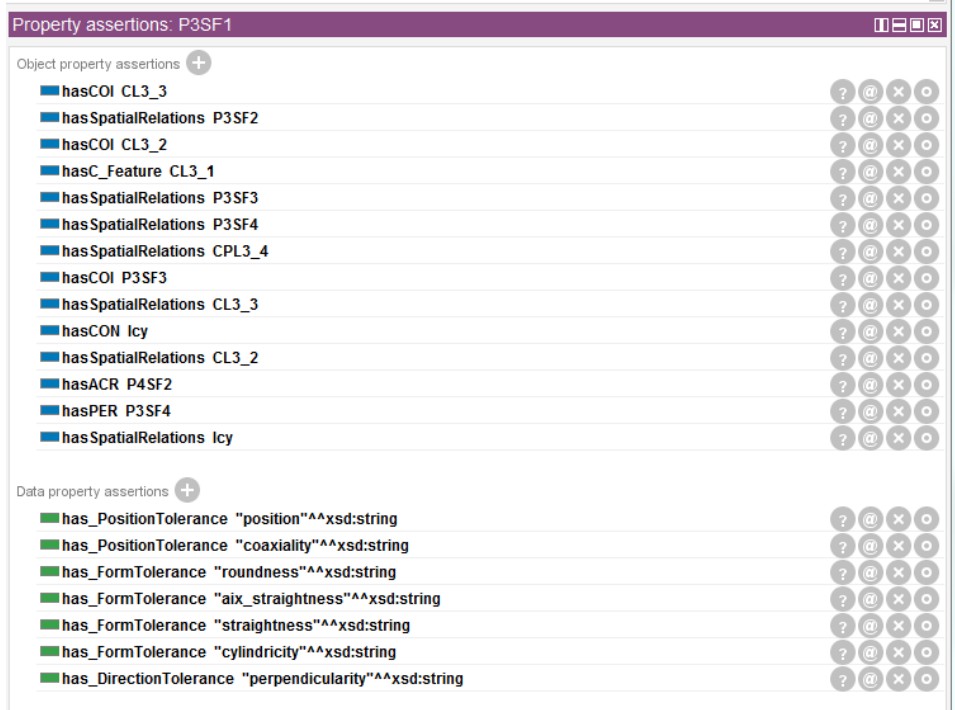

**Figure 13.** Recommended tolerance types for P$_3$SF$_1$.

### 5.2.3. Automatic Screening of AT Types

1. Constructing the set of tolerance type assertions TABOX for each assembly face of Part$_3$. According to the reasoning in Section 5.2.2, the tolerance types of each AFS P$_3$SF$_1$, P$_3$SF$_2$, P$_3$SF$_3$, and P$_3$SF$_4$ of part Part$_3$ were obtained. The selection tolerance type assertion set As was constructed, as represented by Equation (8).

$$
\begin{aligned}
\text{TAs} = \{ &\text{Perpendicularity(Per3\_1), Perpendicularity(Per3\_2), Perpendicularity(Per3\_3),} \\
&\text{Perpendicularity(Per3\_1), Cylindrictity(Cly3\_1), Cylindrictity(Cly3\_2), Cylindrictity(Cly3\_3),} \\
&\text{Flatness(Fla3\_4), Roundness(Rou3\_1), Roundness(Rou3\_2), Roundness(Rou3\_3),} \\
&\text{Straightness(Str3\_1), Straightness(Str3\_2), Straightness(Str3\_3), Straightness(Str3\_4),} \\
&\text{CircularRunOut(CirRunOut3\_2), CircularRunOut(CirRunOut3\_3), CircularRunOut(CirRunOut3\_4),} \\
&\text{TotalRunOut(ToRunOut3\_2), TotalRunOut(ToRunOut3\_3), TotalRunOut(ToRunOut3\_4),} \\
&\text{Coaxianlity(Coa3\_1), Coaxianlity(Coa3\_2), Coaxianlity(Coa3\_3),} \\
&\text{AixStraightness(AixStr3\_1), AixStraightness(AixStr3\_2), AixStraightness(AixStr3\_3),} \\
&\text{Position(Pos3\_1), Position(Pos3\_1), Position(Pos3\_2), Position(Pos3\_3), Position(Pos3\_4)} \}.
\end{aligned}
\tag{8}
$$

2. Setting of the control parameters DOF vector $V_0$. Based on the geometric functional requirements of the assembly and the coordinate system determined in step 2 of Section 5.2.1, we instantiated tolerance-zone DOFs under the DOF class in the tolerance specification design knowledge ontology library. Based on this, the common parameter DOF determination rule and the control parameter DOF determination rule

proposed in Section 4.5.2 were combined to infer the control parameter DOF vector $V_0$ (the instantiation name is represented by CPDF). The data properties of instance CPDF are shown in Figure 14. The control parameter DOF vector $V_0 = (0\ 0\ 0\ 1\ 1\ 0)$ can be obtained from the data attributes of CPDF.

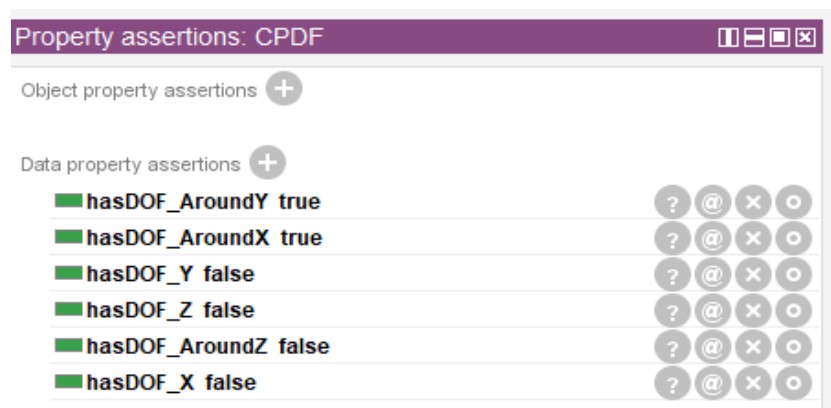

**Figure 14.** Data attributes of CPDF vector for control parameter DOF.

3.  Determination of comparative DOFs for different tolerance types. Using the comparative DOF calculation inference rule proposed in Section 4.5.2, the comparative DOFs for the different tolerance types on the AFSs of the work piece Part$_3$ were inferred. The AFS $P_3S_1$ of the work piece Part$_3$ had multiple recommended tolerance types, and coaxiality tolerance was one of them. Example B_ Coa3 is the comparative DOF for instance Coa3. After reasoning through the SWRL rule for comparing DOFs, the attribute values of the B_Coa3_1 instance were obtained, as shown in Figure 15.

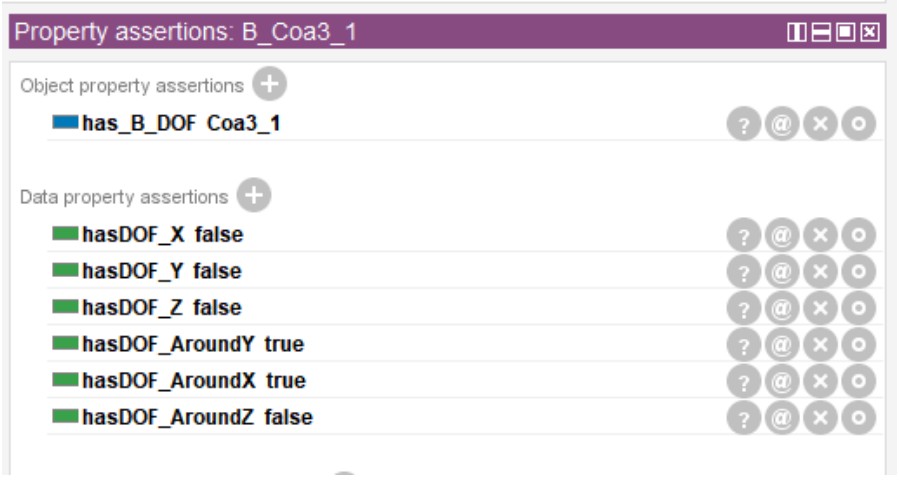

**Figure 15.** The attribute values of B_Coa3_1.

4.  Screening of tolerance types. Using the screening rules proposed in Section 4.5.2, the screening of inferred tolerances for all the feature faces of part Part$_3$ was completed, and the results are shown in Table 5.

**Table 5.** Tolerance after screening of part $Part_3$.

| | ASF [1] | In the Plan | In Any Direction | ○ | ⌀̸ | ↗ | ↗↗ | ⊕ | ◎ | // | ⊥ | ▱ |
|---|---|:---:|:---:|:---:|:---:|:---:|:---:|:---:|:---:|:---:|:---:|:---:|
| Selected tolerances | $P_3S_1$ | √ | | | √ | | √ | √ | √ | | √ | |
| | $P_3S_2$ | √ | | | √ | | √ | √ | √ | | √ | |
| | $P_3S_3$ | √ | | | √ | | √ | √ | √ | | √ | |
| | $P_3S_4$ | | | | | | | √ | √ | | | √ | √ |

### 5.2.4. Detailed Design of Tolerance Specifications

The tolerance types obtained through the screening all met the geometric functional requirements of the product. But, when completing the final detailed tolerance specification design of part $Part_3$, further manual screening of tolerance types, adding tolerance material condition symbols and tolerance domain feature symbols were required [23,32]. The benchmark conditions for the directional positional tolerances were simultaneously selected. The marked tolerance of the final part $Part_3$ is shown in Figure 16.

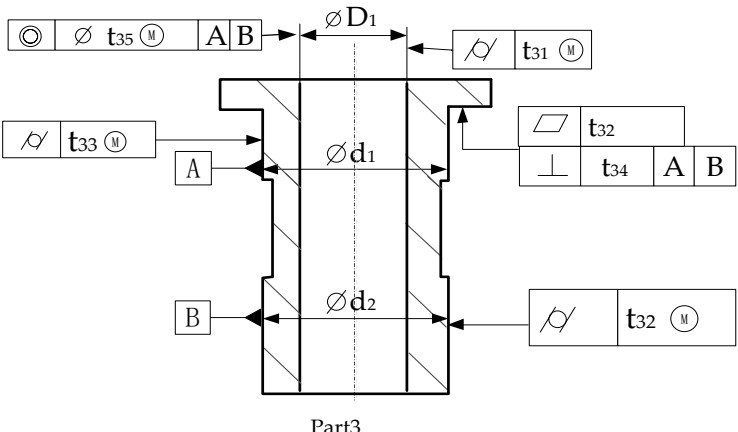

Part3

**Figure 16.** Tolerance marking for part $Part_3$.

## 6. Discussion

In this paper, a semantic representation model of tolerance information based on tolerance-zone DOFs and a semantic model of AT meta-ontology based on tolerance-zone DOFs were proposed. On this basis, a tolerance specification design method combining tolerance-zone DOFs and ontology was constructed, which is able to automatically filter assembly geometric tolerance types. This section compares the method proposed in this paper with the two studies on ontology-based tolerance design and tolerance type screening described in Section 2, as follows.

From the tolerance specification design point of view, our proposed method combines tolerance-zone DOFs with the hierarchical assembly tolerance information representation model, which improves the assembly tolerance information representation model, realises the screening of tolerance types and meets the geometric functional requirements of the product.

From the automatic screening of tolerance types, we constructed a semantic representation model of tolerance information containing tolerance-band freedom. On this basis, the semantic model of assembly tolerance meta-ontology based on tolerance-band freedom ontology was constructed, which can be well integrated with OWL-based ontology technology combined technology integration. For this reason, this proposed approach greatly supports semantic interoperability in assembly tolerance specification design, and the field of tolerance specification design using heterogeneous CAD systems is expanded. In contrast, the automatic tolerance generation method based on assembly positioning

constraints proposed in the literature [27–29], which firstly recommends positional degree tolerance and, on this basis, adds directional tolerance or replaces the positional degree with directional tolerance according to the type of constraints on the assembly feature faces, does not facilitate automatic screening in combination with OWL ontology technology.

From the results of the case validation, the improved tolerance specification design method outlined in this paper achieved the effect of using the tolerance-band degrees of freedom to screen the tolerance types and achieved the purpose of automatically generating tolerance types and a small number of recommended tolerances for the elements of the assembly features of the workpiece. From the case in this paper, the recommended tolerances to be obtained for each assembly feature of part $Part_3$ using the Qin [35] method are shown in Table 6. For example, the cylindrical assembly feature elements $P_3S_2$ and $P_3S_3$ of part $Part_3$ are recommended to obtain 10 kinds of shape and position tolerances, and through the design method of the automatic screening of tolerance types proposed by us, 6 kinds of tolerances were obtained; the planar assembly feature element $P_3S_4$ is recommended to obtain 6 kinds of tolerance types, and after the automatic screening, 4 kinds of tolerances were obtained.

**Table 6.** Automatically generated tolerance types for part $Part_3$ in case of different methods.

| | ASF [1] | — | | ○ | ⌭ | ↗ | ⤢ | ⊕ | ◎ | // | ⊥ | ▱ |
|---|---|---|---|---|---|---|---|---|---|---|---|---|
| | | In the Plan | In Any Direction | | | | | | | | | |
| Qin [35] method | $P_3S_1$ | | ✓ | ✓ | ✓ | | | ✓ | ✓ | ✓ | ✓ | |
| | $P_3S_2$ | ✓ | ✓ | ✓ | ✓ | ✓ | ✓ | ✓ | ✓ | ✓ | ✓ | |
| | $P_3S_3$ | ✓ | ✓ | ✓ | ✓ | ✓ | ✓ | ✓ | ✓ | ✓ | ✓ | |
| | $P_3S_4$ | ✓ | | | | ✓ | ✓ | ✓ | | | ✓ | ✓ |
| Our method | $P_3S_1$ | | ✓ | | ✓ | | | ✓ | ✓ | ✓ | ✓ | |
| | $P_3S_2$ | | ✓ | | ✓ | | | ✓ | ✓ | ✓ | ✓ | |
| | $P_3S_3$ | | ✓ | | ✓ | | | ✓ | ✓ | ✓ | ✓ | |
| | $P_3S_4$ | | | | | | ✓ | ✓ | | | ✓ | ✓ |

## 7. Conclusions

Our proposed method improves the assembly tolerance information representation model by incorporating tolerance-band DOF information into the hierarchical assembly tolerance representation model for the first time. Meanwhile, the vector values of tolerance-band degree-of-freedom vectors in different coordinate systems were defined using matrix expressions, which provide methodological guidance for constructing a TABox (tolerance type example assertion set) for screening tolerance types and constructing inference rules.

Our proposed method also combines a semantic model of geometric tolerance information containing tolerance-band degrees of freedom with an improved hierarchical assembly tolerance information representation model to establish a meta-ontology model of assembly tolerance oriented to filtering tolerance types. Based on the meta-ontology model, a conceptual terminology set oriented to intelligently filtering tolerance types was constructed. On this basis, the ontological framework of a tolerance specification design domain for filtering tolerance types using tolerance-band degrees of freedom was obtained.

In summary, the tolerance specification design method based on tolerance-zone DOFs and ontology is a technology that can automatically generate tolerance types and recommend fewer tolerances. It embodies the advantage of ontology technology, which supports semantic interoperability and enables information exchange between heterogeneous industrial CAD systems. At the same time, the advantage of using tolerance-zone DOFs to screen tolerance types is they it can screen out tolerance types that meet the geometric functional requirements of the assembly. The above methods can benefit mechanical designers and improve the efficiency and reliability of tolerance specification design. However, this method still has shortcomings, as it does not consider the automatic generation and screening of reference standards, which is the direction of the next stage of research.

**Author Contributions:** Conceptualisation, G.L. and M.H.; methodology, G.L. and M.H.; validation, G.L., W.S. and M.H.; formal analysis, G.L.; investigation, G.L.; resources, W.S.; data curation, G.L.; writing—original draft preparation, G.L.; writing—review and editing, G.L. and M.H.; visualisation, M.H.; supervision, W.S.; project administration, G.L.; funding acquisition, M.H. and G.L. All authors have read and agreed to the published version of the manuscript.

**Funding:** This work was supported by the National Natural Science Foundation of China (No. 52165064) and was supported by the Project on Enhancement of Basic Research Ability of Young and Middle-aged Teachers in Universities and Colleges of Guangxi (No. 20020KY31008).

**Institutional Review Board Statement:** Not applicable.

**Informed Consent Statement:** Not applicable.

**Data Availability Statement:** Data are contained within the article.

**Acknowledgments:** We would like to acknowledge Meifa Huang for her guidance, revisions and help in writing the manuscript, and Wenbo Su for providing technical information for writing the paper and proofreading the paper.

**Conflicts of Interest:** The authors declare no conflicts of interest.

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
