# Peer review of "A Tolerance Specification Automatic Design Method for Screening Geometric Tolerance Types"

_applsci, doi:10.3390/app14031302_

Round 1

Reviewer 1 Report

Comments and Suggestions for Authors

“Reviewer’s comments”

The submitted manuscript demonstrates a commendable integration of ontology and tolerance-zone DOFs for tolerance specification design. It is recommended that the authors undertake extensive revisions to address the following concerns and enhance the manuscript's coherence, clarity, and overall contribution to the field before considering it for publication.

Required modifications:

- The end of the introduction paragraph needs a more precise and concise explanation of the novelty and uniqueness of the paper.

- Clarify how the integration of geometric tolerance information using tolerance-zone DOFs approach enhances existing methods and what specific challenges it addresses in tolerance specification.

- Elaborate more on how the Protégé tool is used in the representation process.

- In order to gain a better understanding of the application, it is necessary to obtain additional details on the methodology employed. Furthermore, it is important to be aware of any potential limitations associated with the applied approach in order to ensure accurate and reliable results.

- The manuscript would benefit from revising the language to improve clarity and readability.

- Consistency and Formatting Recommendations:

Maintain a consistent style for citing figures throughout the manuscript. Either use "Fig. 1" consistently or "Figure 1" consistently.

Standardize the spacing before and after citations. Ensure uniformity by either consistently including a space before and after citations or omitting spaces altogether.

Verify that all abbreviations and acronyms used in the manuscript are explicitly defined before their initial mention in the text.

Comments on the Quality of English Language

Minor editing of English language required.

Author Response

Please see the attchement.

Reviewer 2 Report

Comments and Suggestions for Authors

The manuscript's authors, “A tolerance specification automatic design method for screening geometric tolerance types”, presented a methodology for designing tolerance specifications. The topic of the manuscript is relevant to scientific and manufacturing communities.

However, some aspects can be improved:

Page 2, paragraph 2, line 1

1.       Please explain the AT abbreviation.

2.       In the introduction section, it is recommended to specify the paper's main idea. It was mentioned that the current paper presented the methodology for designing tolerance specifications, the necessity or importance of developing such a methodology needed to be logically proved.

3.        Considering more temporary articles on the topic is highly recommended. So far, more than 68% of references are older than 5 years.

4.        Figure 1. Please correct the word “assembly” from the capital letter.

5.        In paragraph 4.1, in the sentence, “Reference [14] defines the DOF of tolerance zones, and on this basis, the selection of tolerance types for component assembly feature features is achieved.”  There is probably a typo. Please correct.

6.       Please explain SWERL abbreviation and how it differs from SWRL?

7.       Please discuss the results achieved in the discussion and conclusions section. Compare the advantages and disadvantages of the methodology proposed in the current paper and earlier studies.

8.       Please draw out more detailed conclusions. So far, thoughts in the last paragraph of the Discussion and Conclusion Section are very brief for such a detailed methodological paper. In conclusions section, it is proposed to follow the paper structure and draw at least one outcome from each chapter.

Reviewer 3 Report

Comments and Suggestions for Authors

It is a well-written paper, on a high-level field that aims to develop a method for the automatic selection of assembly tolerance types in a faster and more precise way.

Reading the work was quite difficult, on the one hand because of some long sentences, on the other hand due to a large condensation of information, that actually hides research and models that the authors carried out and were presented in some others papers.

  Analyzing the literature, I found several papers done by this research team, which helped me to a better understanding of this one.

These are:

1. Guanghao Liu 1,2,Meifa Huang 1,* and Leilei Chen, Optimization Method of Assembly Tolerance Types Based on Degree of Freedom

2. Zhiguo Peng a b, Meifa Huang a b, Yanru Zhong a, Leilei Chen a b, Guanghao Liu, A new method for interoperability and conformance checking of product manufacturing information

3. Zhiguo Peng, Meifa Huang, Yanru Zhong, Leilei Chen, Guanghao Liu, Enhanced semantic representation of coaxiality with double material requirements

4. Yuchu Qin, Wenlong Lu, Q. Qi, Tukun Li, Meifa Huang, P. Scott, X. Jiang. Explicitly representing the semantics of composite positional tolerance for patterns of holes

Maybe there are some others papers too.

I mention that these papers are not listed in the bibliography, perhaps to avoid self-citations. However, without going through them, especially paper 1 from the above list, it is difficult to understand this paper. I propose to include them in the Bibliography as well.

This work has a high level of mathematic modelling, required for the necessity of CAD systems development. In order to facilitate the reading and understanding of this work, please create a list of abbreviations, with an explanation of their meanings.

Many aspects of the model are also presented in other papers of this team. However, I do not consider this plagiarism because in this work there is a context related to software development.

  Please improve the chapter: "Implementation and Examples", in the following way.

 The purpose of the work, is to develop a method" capable to "automatically generate tolerance types and recommend fewer tolerances". So, you can easily generate the tolerances of various assemblies based on this model.  Near the example you presented, which is identical to the one from paper 1 from the above list (it is good to keep the existing example too because it is more technically presented in paper 1), please analyze some simpler examples and emphasize the advantages of this design. These simpler examples would allow a deeper understanding of the design method, you propose.

  Regarding the analyzed example, to emphasize the advantage of the method you wrote: "As can be seen from Table 4: such as the cylinder assembly feature element P3S2 of work piece Part3, using Qin [31] method, we? recommend 10 types of geometric tolerances, and our proposed method obtains 6 types of tolerances. The tolerance types for manual intervention decision are reduced by 4 compared to Qin [31] method, All the tolerance types obtained from the screening satisfy the need to control the geometric function of the assembly. For example, it avoids controlling shape tolerances with a single roundness or straightness element; in the examples in this paper, it also avoids choosing a combination of roundness and straightness to control shape errors in the cylinder.”

 We don't have Qin's results in front of us... It is not at all clear how your results are compared to Qin's results... You can also show Qin's results and emphasize the advantage of your method.

   The conclusions are not detailed and not clear enough to provide an understanding of the research. Please describe shortly the modules (stages, levels) of the method and the way in which information passes from one level to another. It is necessary to have a succinct description of the stages described in chapter 5, "Implementation and Examples". So, please elaborate and improve the conclusions.

 Figures 2 and 3 have grammatical errors (Tolerance !!!)

  I appreciate the team and the research they are doing and I wish them much success in the future.

Round 2

Reviewer 1 Report

Comments and Suggestions for Authors

I recommend accepting the paper.